# MONOTONE DEEP BOLTZMANN MACHINES

## ABSTRACT

Deep Boltzmann machines refer to deep multi-layered probabilistic models, governed by a pairwise energy function that describes the likelihood of all variables in the network. Due to the difficulty of inference in such systems, they have given way largely to *restricted* deep Boltzmann machines (which do not permit intra-layer or skip connections). In this paper, we propose a class of model that allows for *exact, efficient* mean-field inference and learning in *general* deep Boltzmann machines. To do so, we use the tools of the recently proposed monotone Deep Equilibrium (DEQ) Model, an implicit-depth deep network that always guarantees the existence and uniqueness of its fixed points. We show that, for a class of general deep Boltzmann machine, the mean-field fixed point can be considered as the equivalent fixed point of a monotone DEQ, which gives us a recipe for deriving an efficient mean-field inference procedure with global convergence guarantees. We apply this approach to simple deep convolutional Boltzmann architectures and demonstrate that it allows for tasks such as the joint completion and classification of images, all within a single deep probabilistic setting.

## 1 INTRODUCTION

This paper considers (deep) Boltzmann machines, which are pairwise energy-based probabilistic models. Theses models specify a joint distribution over variables $\mathbf{x}$ given by the density

$$p(\mathbf{x}) \propto \exp\left(\sum_{(i,j)\in E} x_i^\top \Phi_{ij} x_j + \sum_{i=1}^n b_i^\top x_i\right), \tag{1}$$

where each $x_{1:n}$ denotes a discrete random variable over $k_i$ possible values, represented as a one-hot encoding $x_i \in \{0,1\}^{k_i}$; $E$ denotes the set of edges in the model; $\Phi_{i,j} \in \mathbb{R}^{k_i \times k_j}$ represents pairwise potential; and $b_i \in \mathbb{R}^{k_i}$ represents unary potential. Depending on context, these models are typically referred to as pairwise Markov random fields (MRFs) (Koller & Friedman, 2009), or (potentially deep) Boltzmann machines (Goodfellow et al., 2016; Salakhutdinov & Hinton, 2009; Hinton, 2002). In the above setting each $x_i$ may represent an observed or unobserved value, and there can be substantial structure within the variables; for instance, the collection of variables $\mathbf{x}$ may (and indeed will, in the main settings we consider in this paper) consist of several different "layers" in a joint convolutional structure, leading to the deep convolutional Boltzmann machine (Norouzi et al., 2009).

In this paper, we propose a new parameterization and algorithmic approach to approximate inference of these probabilistic models. There are two main contributions: First, we define a generic parameterization of the pairwise kernel function $\Phi$, that can represent a general Boltzmann machine. Our parametrization is flexible enough to incorporate almost all operators and network topology, including fully-connected layers, convolution operators, and skip-connections, etc. Through the lens of the recently developed monotone DEQ (Winston & Kolter, 2020), we constraint $\Phi$ to satisfy certain monotonicity conditions through training and inference; this thereby assures that the mean-field approximation will always have a *unique, globally-optimal* fixed point under this parameterization. Second, although previous works (Krähenbühl & Koltun, 2013; Baqué et al., 2016) have made approaches to parallel mean-field updates, they either require strong conditions on $\phi$, or fail to converge to the true mean-field distribution. We provide a properly-damped mean-field update method, based upon a generic proximal operator, which is guaranteed to converge to the mean-field fixed point, even if applied in parallel to all random variables simultaneously. Although there is no exact

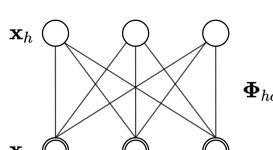
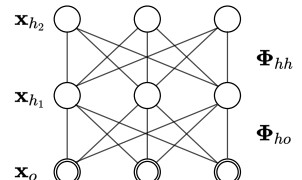
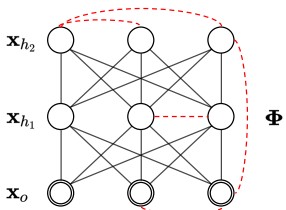

Figure 1: Neural network topology of different Boltzmann machines. The general case is a complete graph. Our proposed parameterization is equivalent to a general Boltzmann machine. Technically in our setting the $\Phi_{oo}$ connections exist, but they are not updated due to the input injection.

closed form solution of this proximal operator, we derive a very efficient Newton-based implementation. To estimate the parameters of our model, we follow the marginal-based loss minimization approach in Krähenbühl & Koltun (2013); Domke (2013), where the objective is to directly maximize the likelihood induced by the approximated mean-field distribution. Such an approach grants us the ability to update parameters via taking gradient steps of the proposed mean-field iterations.

With the proposed approaches, we perform both learning and inference for a deep convolutional, multi-resolution Boltzmann machine, and apply the network to model MNIST and CIFAR-10 pixels and their classes conditioned on partially observed images. Such joint probabilistic modelling allows us to simultaneously impute missing pixels and predict the class. While these are naturally a small-scale problem, we emphasize that performing joint probabilistic inference over a complete model of this type is a relatively high-dimensional task as far as traditional mean-field inference is concerned. We also compare our inference method to previous ones and demonstrate different convergence properties of each. We conclude by highlighting limitations and directions for future work with the method.

## 2  BACKGROUND AND RELATED WORK

This paper builds upon three main avenues of work: 1) deep equilibrium models, especially their convergent version, the monotone DEQ; 2) the broad topic of energy-based deep model and Boltzmann machines in particular; and 3) work on concave potentials and parallel methods for mean-field inference. We discuss each of these below.

**Equilibrium models and their provable convergence**    The DEQ model was first proposed by Bai et al. (2019). Based on the observation that a neural network $z^{t+1} = \sigma(Wz_t + Ux + b)$ with input injection $x$ usually converges to a fixed point, they modeled an effectively infinite-depth network with input injection directly via its fixed point: $z^* = \sigma(Wz^* + Ux + b)$. Its backpropagation is done through the implicit function theorem and only requires constant memory. Bai et al. (2020) also showed that the multiscale DEQ models achieve near state-of-the-art performances on many large-scale tasks. Winston & Kolter (2020) later presented a parametrization of the DEQ (denoted as monDEQ) that guarantees provable convergence to a unique fixed point, using monotone operator theory. Specifically, they parameterize $W$ in a way that $I - W \succeq mI$ (called $m$-strongly monotone) is always satisfied during training for some $m > 0$; they convert nonlinearities into proximal operators (which include ReLU, tanh, etc.), and show that using existing splitting methods like *forward-backward* and *Peaceman-Rachford* can provably find the unique fixed point.

**Markov random field (MRF) and its variants**    MRF is a form of energy-based model, which model joint probabilities of the form $p_\theta(x) = \exp(-E_\theta(x))/Z_\theta$ for an energy function $E_\theta$. A common type of MRF is the Boltzmann machine, the most successful variant of which is the restricted Boltzmann machines (RBM) (Hinton, 2002) and its deep (multi-layer) variant (Salakhutdinov & Hinton, 2009). Particularly, RBMs define $E_\theta(v, h) = -a^\top v - b^\top h - v^\top Wh$, where $\theta = \{W, a, b\}$, $v$ is the set of visible variables, and $h$ is the set of latent variables. It is usually trained using the

contrastive-divergence algorithm, and its inference can be done efficiently by a block mean-field approximation. However, a particular restriction of RBMs is that there can be no intra-layer connections, that is, each variable in $v$ (resp. $h$) is independent conditioned on $h$ (resp. $v$). A deep RBM allows different layers of hidden nodes, but there cannot be intra-layer connections. By contrast, our formulation allows intra-layer connections and is therefore is more expressive in this respect. See fig. 1 for the network topology of RBM, deep RBM, and general BM (we also use the term general deep BM interchangeably to emphasize the existence of deep structure). Wu et al. (2016) proposed a deep parameterization of MRF, but their setting only considers a grid of hidden variables $h$ and the connections among hidden units are restricted to the neighboring nodes. Therefore, it is a special case of our parameterization (although their learning algorithm is orthogonal to ours). Numerous works also try to combine deep neural networks with conditional random fields (CRF) (Krähenbühl & Koltun, 2013; Zheng et al., 2015; Schwartz et al., 2017) These models either train a pre-determined kernel as an RNN or use neural networks for producing either inputs or parameters of their CRFs.

**Parallel and convergent mean-field**   It is well-known that mean-field updates converge locally using a coordinate ascent algorithm (Blei et al., 2017). However, local convergence is only guaranteed if the update is applied sequentially. Nonetheless, several works have proposed techniques to parallelize updates. Krähenbühl & Koltun (2013) proposed a concave-convex procedure (CCCP) to minimize the KL divergence between the true distribution and the mean-field variational family. To achieve efficient inference, they use a concave approximation to the pairwise kernel, and their fast update rule only converges if the kernel function is concave. Later, Baqué et al. (2016) derived a similar parallel damped forward iteration to ours that provably converges without the concave potential constraint. However, unlike our approach, they do not use a parameterization which ensures a global mean-field optimum, and their algorithm therefore may not converge to the actual fixed point of the mean-field updates. This is because Baqué et al. (2016) used the $\text{prox}_f^1$ proximal operator (described below), whereas we derive the $\text{prox}_f^\alpha$ operator to guarantee global convergence when doing mean-field updates in parallel. What's more, Baqué et al. (2016) focused only on inference over prescribed potentials, and not on training the (fully parameterized) potentials as we do here. Lê-Huu & Alahari (2021) brought up a generalized Frank-Wolfe based framework for mean-field updates which include the methods proposed by Baqué et al. (2016); Krähenbühl & Koltun (2013). Their results only guarantee global convergence to a local optimal.

## 3   MONOTONE DEEP BOLTZMANN MACHINES AND APPROXIMATE INFERENCE

In this section, we present the main technical contributions of this work. We begin by presenting a parameterization of the pairwise potential in a Boltzmann machine that guarantees the monotonicity condition. We then illustrate the connection between a (joint) mean-field inference fixed point and the fixed point of our monotone Boltzmann machine and discuss how deep structured networks can be implemented in this form practically; this establishes that, under the monotonicity conditions on $\Phi$, there exists a unique globally-optimal mean-field fixed point. Finally, we present an efficient parallel method for computing this mean-field fixed point, again motivated by the machinery of monotone DEQs and operator splitting methods.

### 3.1   A MONOTONE PARAMETERIZATION OF GENERAL BOLTZMANN MACHINES

In this section, we show how to parameterize our probabilistic model in a way that the pairwise potentials satisfy $I - \Phi \succeq mI$, which will be used later to show the existence of a unique mean-field fixed point. Additionally, since $\Phi$ defines a graphical model that has no self-loop, we further require $\Phi$ to be a *block hollow* matrix (that is, the $k_i \times k_i$ diagonal blocks corresponding to each variable must be zero). While both these conditions on $\Phi$ are convex constraints, in practice it would be extremely difficult to project a generic set of weights onto this constraint set under an ordinary parameterization of the network.

Thus, we instead advocate for a *non-convex* parameterization of the network weights, but one which guarantees that the monotonicity condition is always satisfied, without any constraint on the weights in the parameterization. Specifically, define the block matrix

$$\boldsymbol{A} = [\begin{array}{cccc} A_1 & A_2 & \cdots & A_n \end{array}] \tag{2}$$

with $A_i \in \mathbb{R}^{d \times k_i}$ matrices for each variables, and where $d$ can be some arbitrarily chosen dimension. Then let $\hat{A}_i$ be a spectrally-normalized version of $A_i$

$$\hat{A}_i = A_i \cdot \min\{\sqrt{1-m}/\|A_i\|_2, 1\} \tag{3}$$

i.e., a version of $A_i$ normalized such that its largest singular value is at most $\sqrt{1-m}$ (note that we can compute the spectral norm of $A_i$ as $\|A_i\|_2 = \|A_i^T A_i\|_2^{1/2}$, which involves computing the singular values of only a $k_i \times k_i$ matrix, and thus is very fast in practice). We define the $\hat{A}$ matrix analogously as the block version of these normalized matrices.

Then we propose to parameterize $\mathbf{\Phi}$ as

$$\mathbf{\Phi} = \text{blkdiag}(\hat{A}^T \hat{A}) - \hat{A}^T \hat{A} \tag{4}$$

where $\text{blkdiag}$ denotes the block-diagonal portion of the matrix along the $k_i \times k_i$ block. Put another way, this parameterizes $\mathbf{\Phi}$ as

$$\Phi_{ij} = \begin{cases} -\hat{A}_i^T \hat{A}_j & \text{if } i \neq j, \\ 0 & \text{if } i = j. \end{cases} \tag{5}$$

As the following simple theorem shows, this parameterization guarantees both hollowness of the $\mathbf{\Phi}$ matrix and monotonicity of $I - \mathbf{\Phi}$, for any value of the $A$ matrix.

**Theorem 3.1.** *For any choice of parameters $A$, under the parametrization equation 4 above, we have that 1) $\Phi_{ii} = 0$ for all $i = 1, \ldots, n$, and 2) $I - \mathbf{\Phi} \succeq mI$.*

*Proof.* Block hollowness of the matrix follows immediately from construction. To establish monotonicity, note that

$$\begin{aligned} I - \mathbf{\Phi} \succeq mI &\iff I + \hat{A}^T \hat{A} - \text{blkdiag}(\hat{A}^T \hat{A}) \succeq mI \\ &\impliedby I - \text{blkdiag}(\hat{A}^T \hat{A}) \succeq mI \\ &\iff I - \hat{A}_i^T \hat{A}_i \succeq mI, \ \forall i \\ &\iff \|\hat{A}_i\|_2 \leq \sqrt{1-m}, \ \forall i. \end{aligned} \tag{6}$$

This last property always holds by the construction of $\hat{A}_i$. $\qquad\square$

## 3.2 Mean-field inference as a monotone DEQ

In this section, we formally present how to formulate the mean-field inference as a DEQ update. Recall from before that we are modelling a distribution of the form eq. (1). We are interested in approximating the conditional distribution $p(\mathbf{x}_h | \mathbf{x}_o)$, where $o$ and $h$ denote the observed and hidden variables respectively, with a factored distribution $q(\mathbf{x}_h) = \prod_{i \in h} q_i(x_i)$. Here, the standard mean-field updates (which minimize the KL divergence between $q(\mathbf{x}_h)$ and $p(\mathbf{x}_h | \mathbf{x}_o)$ over the single distribution $q_i(x_i)$) are given by the following equation,

$$q_i(x_i) := \text{softmax} \left( \sum_{j:(i,j) \in E} \Phi_{ij} q_j(x_j) + b_i \right) \tag{7}$$

where overloading notation slightly, we let $q_j(x_j)$ denote a one-hot encoding of the observed value for any $j \in o$ (see e.g., Koller & Friedman (2009) for a full derivation).

The essence of the above updates is a characterization of the joint fixed point to mean-field inference. For simplicity of notation, defining $\boldsymbol{q} = [q_1(x_1) \quad q_2(x_2) \quad \ldots]^T$.

We see that $\boldsymbol{q}_h$ is a joint fixed point of all the mean-field updates if and only if

$$\boldsymbol{q}_h = \text{softmax}\left(\mathbf{\Phi}_{hh} \boldsymbol{q}_h + \mathbf{\Phi}_{ho} \mathbf{x}_o + \boldsymbol{b}_h\right) \tag{8}$$

where $\mathbf{x}_o$ analogously denotes the stacked one-hot encoding of the observed variables.

We briefly recall the monotone DEQ framework of Winston & Kolter (2020). Given input vector $\mathbf{x}$, a monotone DEQ computes the fixed point $\boldsymbol{z}^\star(\mathbf{x})$ that satisfies the equilibrium equation

$$\boldsymbol{z}^\star(\mathbf{x}) = \sigma(W \boldsymbol{z}^\star(\mathbf{x}) + U\mathbf{x} + b). \tag{9}$$

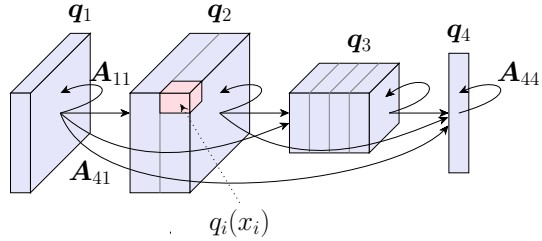

Figure 2: Illustration of a possible deep convolutional Boltzmann machine, where the monotonicity structure can still be enforced.

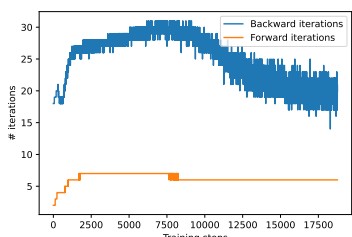

Figure 3: Convergence of the forward-backward splitting.

Winston & Kolter (2020) showed that if: 1) $\sigma$ is given by a proximal operator[1] $\sigma(x) = \operatorname{prox}_f^1(x)$ for some convex closed proper (CCP) $f$, and 2) if we have the monotonicity condition $I - W \succeq mI$ (in the positive semidefinite sense) for some $m > 0$, then for any $\mathbf{x}$ there exists a unique fixed point $z^\star(\mathbf{x})$, which can be computed through standard operator splitting methods, such as forward-backward splitting.

We now state our main claim of this subsection, that under certain conditions the mean-field fixed point can be viewed as the fixed point of an analogous DEQ. This is formalized in the following proposition.

**Proposition 3.1.** *Suppose that the pairwise kernel $\mathbf{\Phi}$ satisfies $I - \mathbf{\Phi} \succeq mI$ [2] for $m > 0$. Then the mean-field fixed point*

$$q_h = \operatorname{softmax}\left(\mathbf{\Phi}_{hh}q_h + \mathbf{\Phi}_{ho}\mathbf{x}_o + \boldsymbol{b}_h\right) \tag{10}$$

*corresponds to the fixed point of a monotone DEQ model. Specifically, this implies that for any $\mathbf{x}_o$, there exists a unique, globally-optimal fixed point of the mean-field distribution $q_h$.*

As the monotonicity condition of the monotone DEQ is assumed in the proposition, the proof of the proposition rests entirely in showing that the softmax operator is given by $\operatorname{prox}_f^1$ for some CCP $f$. Specifically, as shown in (Krähenbühl & Koltun, 2013), this is the case for

$$f(z) = \sum_i z_i \log z_i - \frac{1}{2}\|z\|_2^2 + I\left\{\sum_i z_i = 1, \; z_i \geq 0\right\} \tag{11}$$

i.e., the restriction of the entropy minus squared norm to the simplex (note that even though we are *subtracting* a squared norm term it is straightforward to show that this function is convex, since the second derivatives are given by $1/z_i - 1$, which is always non-negative over its domain).

### 3.3 PRACTICAL CONSIDERATIONS WHEN MODELLING MONOTONE BOLTZMANN MACHINES

The construction in section 3.1 guarantees monotonicity of the resulting pairwise probabilistic model. However, instantiating the model in practice, where the variables represent hidden units of a deep architecture (i.e., representing multi-channel image tensors with pairwise potentials defined by convolutional operators), requires substantial subtlety and care in implementation. In this setting, we do not want to actually represent $\boldsymbol{A}$ explicitly, but rather determine a method for *multiplying* $\boldsymbol{A}\boldsymbol{v}$ and $\boldsymbol{A}^T\boldsymbol{v}$ for some vector $\boldsymbol{v}$ (as we see in section 3.2, this is all that is required for the parallel mean-field inference method we propose). This means that certain blocks of $\boldsymbol{A}$ are typically parameterized as convolutional layers, with convolution and transposed convolution operators as the main units of computation.

More specifically, we typically want to *partition* the full set of hidden units into some $K$ distinct sets

$$q = \begin{bmatrix} \boldsymbol{q}_1 & \boldsymbol{q}_2 & \cdots & \boldsymbol{q}_K \end{bmatrix}^T \tag{12}$$

---

[1]A proximal operator is defined by $\operatorname{prox}_f^\alpha(x) = \arg\min_z \frac{1}{2}\|x - z\|^2 + \alpha f(z)$.

[2]Technically speaking, we only need $I - \mathbf{\Phi}_{hh} \succeq mI$, but since we want this to hold for any choice of $h$, we need the condition to apply to the entire $\mathbf{\Phi}$ matrix.

where e.g., $\boldsymbol{q}_i$ would be best represented as a height $\times$ width $\times$ groups $\times$ cardinality tensor (i.e., a collection of a multiple hidden units corresponding to different locations in a typical deep network hidden layer). Note that here $\boldsymbol{q}_i$ is *not* the same as $q_i(x_i)$, but rather the collection of *many* different individual variables. These $\boldsymbol{q}_i$ terms can be related to each other via different operators, and a natural manner of parameterizing $\boldsymbol{A}$ in this case is as an interconnected set of a convolutional or dense operators. To represent the pairwise interactions, we can create a similarly-factored matrix $\boldsymbol{A}$, e.g., one of the form

$$
\boldsymbol{A} = \begin{bmatrix}
\boldsymbol{A}_{11} & 0 & \cdots & 0 \\
\boldsymbol{A}_{21} & \boldsymbol{A}_{22} & \cdots & 0 \\
\vdots & \vdots & \ddots & \vdots \\
\boldsymbol{A}_{K1} & \boldsymbol{A}_{K2} & \cdots & \boldsymbol{A}_{KK}
\end{bmatrix}
\tag{13}
$$

where e.g., $\boldsymbol{A}_{ij}$ is a (possibly strided) convolution mapping between the tensors representing $\boldsymbol{q}_j$ and $\boldsymbol{q}_i$. In this case, we emphasize that $\boldsymbol{A}_{ij}$ is not the kernel matrix that one "slides" along the variables. Instead, $\boldsymbol{A}_{ij}$ is the linear mapping as if we write the convolution as a matrix-matrix multiplication. For example, a 2D convolution with stride 1 can be expressed as a doubly block circulant matrix (the case is more complicated when different striding is allowed). This parametrization is effectively a *general* Boltzmann machine, since each random variable in eq. (12) can interactive with any other variables except for itself. Varying $\boldsymbol{A}_{ij}$, the formulation in eq. (13) is rich enough for any types of architectures including convolutions, fully-connected layers, and skip-connections, etc.

An illustration of one possible network structure is shown in fig. 2. The precise details of how one computes the block diagonal elements of $\boldsymbol{A}^T\boldsymbol{A}$, and how one normalizes the proper diagonal blocks (which, we emphasize, still just requires computing the singular values of matrices whose size is the cardinality of a single $q_i(x_i)$) are somewhat involved, so we defer a complete description to the Appendix (and accompanying code). The larger takeaway message, though, is that *it is possible to parameterize complex convolutional multi-scale Boltzmann machines, all while ensuring monotonicity*.

## 3.4 Efficient parallel solving for the mean-field fixed point

Although the monotonicity of $\boldsymbol{\Phi}$ guarantees the existence of a unique solution, it does not necessarily guarantee that the simple iteration

$$
\boldsymbol{q}_h^{(t)} = \text{softmax}(\boldsymbol{\Phi}_{hh}\boldsymbol{q}_h^{(t-1)} + \boldsymbol{\Phi}_{ho}\boldsymbol{x}_o + \boldsymbol{b}_h)
\tag{14}
$$

will converge to this solution. Instead, to guarantee convergence, one needs to apply the *damped* iteration (see, e.g. (Winston & Kolter, 2020))

$$
\boldsymbol{q}_h^{(t)} = \text{prox}_f^\alpha \left( (1-\alpha)\boldsymbol{q}_h^{(t-1)} + \alpha(\boldsymbol{\Phi}_{hh}\boldsymbol{q}_h^{(t-1)} + \boldsymbol{\Phi}_{ho}\boldsymbol{x}_o + \boldsymbol{b}_h) \right).
\tag{15}
$$

The damped forward-backward iteration converges linearly to the unique fixed point if $\alpha \le 2m/L^2$, assuming $I - \boldsymbol{\Phi}$ is $m$-strongly monotone and $L$-Lipschitz (Ryu & Boyd, 2016). Crucially, this update can be formed *in parallel* over all the variables in the network: we do not require a coordinate descent approach as is typically needed by mean-field inference.

The key issue, though is that while $\text{prox}_f^1(x) = \text{softmax}(x)$ for $f$ defined as in eq. (11), in general this does not hold for $\alpha \ne 1$. Indeed, for $\alpha \ne 1$, there is no closed form solution to the proximal operation, and computing the solution is substantially more involved. Specifically, computing this proximal operator involves solving the optimization problem

$$
\text{prox}_f^\alpha(x) = \underset{z}{\arg\min} \; \frac{1}{2}\|x - z\|_2^2 + \alpha\sum_i z_i \log z_i - \frac{\alpha}{2}\|z\|_2^2
$$
$$
\text{subject to} \; \sum_i z_i = 1, \; z \ge 0.
\tag{16}
$$

The following theorem, proved in the Appendix, characterizes the solution to this problem for $\alpha \in (0, 1)$ (although it is also possible to compute solutions for $\alpha > 1$, this is not needed in practice, as it corresponds to a "negatively damped" update, and it is typically better to simply use the softmax update in such cases).

**Theorem 3.2.** *Given $f$ as defined in eq. (11), $\alpha \in (0, 1)$, and $x \in \mathbb{R}^k$, the proximal operator* $\mathrm{prox}_f^\alpha(x)$ *is given by*

$$\mathrm{prox}_f^\alpha(x)_i = \frac{\alpha}{1-\alpha} W\left(\frac{1-\alpha}{\alpha} \exp\left(\frac{x_i - \alpha + \lambda}{\alpha}\right)\right),$$

*where $\lambda \in \mathbb{R}$ is the unique solution chosen to ensure that the resulting $\sum_i \mathrm{prox}_f^\alpha(x_i) = 1$, and where $W(\cdot)$ is the principal branch of the Lambert $W$ function.*

In practice, however, this is not the most numerically stable method for computing the proximal operator, especially for small $\alpha$, owing to the large term inside the exponential. Computing the proximal operation efficiently is somewhat involved, though briefly, we define the alternative function

$$g(y) = \log \frac{\alpha}{1-\alpha} W\left(\frac{1-\alpha}{\alpha} \exp\left(\frac{y}{\alpha} - 1\right)\right) \tag{17}$$

and show how to directly compute $g(y)$ using Halley's method (note that Halley's method is also the preferred manner to computing the Lambert $W$ function itself numerically (Corless et al., 1996)). Finding the prox operator then requires that we find $\lambda$ such that $\sum_{i=1}^k \exp(g(x_i + \lambda)) = 1$. This can be done via (one-dimensional) root finding with Newton's method, which is guaranteed to always find a solution here, owing to the fact that this function is convex monotonic for $\lambda \in (-\infty, 1)$. We can further compute the gradients of the $g$ function and of the proximal operator itself via implicit differentiation (i.e., we can do it analytically without requiring unrolling the Newton or Halley iteration). We describe the details in the Appendix, and include an efficient PyTorch function implementation in the supplementary material.

**Comparison to Winston & Kolter (2020)**   Although this work uses the same monotonicity constraint as in Winston & Kolter (2020), our result further requires the linear module $\boldsymbol{\Phi}$ to be hollow, and extend their work to the softmax nonlinear operator as well. These extensions introduce significant complications, but also enable us to interpret our network as a probabilistic model, while the network in Winston & Kolter (2020) cannot.

## 3.5 TRAINING CONSIDERATIONS

Finally, we discuss approaches for training these monotone Boltzmann machines, exploiting their efficient approach to mean-field inference. Probabilistic models are typically trained via approximate likelihood maximization, and since the mean-field approximation is based upon a particular likelihood approximation, it may seem most natural to use this same approximation to train parameters. In practice, however, this is often a suboptimal approach. Specifically, because our forward inference procedure ultimately uses mean-field inference, it is better to train the model directly to output the correct marginals, *when running this mean-field procedure*. This is known as a marginal-based loss (Domke, 2013). In the context of monotone Boltzmann machines, this procedure has a particularly convenient form, as it corresponds roughly to the "typical" training of DEQ.

In more detail, suppose we are given a sample $\mathbf{x} \in \mathcal{X}$ (i.e., at training time the entire sample is given), along with a specification of the "observed" and "hidden" sets, $o$ and $h$ respectively. Note that the choice of observed and hidden sets is potentially up to the algorithm designer, and can effectively allows one to train our model in a "self-supervised" fashion, where the goal is to predict some unobserved components from others. In practice, however, one typically wants to design hidden and observed portions congruent with the eventual use of the model: e.g., if one is using the model for classification, then at training time it makes sense for the label to be "hidden" and the input to be "observed."

Given this sample, we first solve the mean-field inference problem to find $\boldsymbol{q}_h^\star(\mathbf{x}_h)$ such that

$$\boldsymbol{q}_h^\star = \mathrm{softmax}\left(\boldsymbol{\Phi}_{hh}\boldsymbol{q}_h^\star + \boldsymbol{\Phi}_{ho}\mathbf{x}_o + \boldsymbol{b}_h\right). \tag{18}$$

For this sample, we know that the true value of the hidden states is given by $\mathbf{x}_h$. Thus, we can apply some loss function $\ell(\boldsymbol{q}_h^\star, \mathbf{x}_h)$ between the prediction and true value, and update parameters of the model $\theta = \{\boldsymbol{A}, \boldsymbol{b}\}$ using their gradients

$$\frac{\partial \ell(\boldsymbol{q}_h^\star, \mathbf{x}_h)}{\partial \theta} = \frac{\partial \ell(\boldsymbol{q}_h^\star, \mathbf{x}_h)}{\partial \boldsymbol{q}_h^\star} \frac{\partial \boldsymbol{q}_h^\star}{\partial \theta} = \frac{\partial \ell(\boldsymbol{q}_h^\star, \mathbf{x}_h)}{\partial \boldsymbol{q}_h^\star} \left(I - \frac{\partial g(\boldsymbol{q}_h^\star)}{\partial \boldsymbol{q}_h^\star}\right)^{-1} \frac{\partial g(\boldsymbol{q}_h^\star)}{\partial \theta} \tag{19}$$

with $g(\boldsymbol{q}_h^\star) \equiv \text{prox}_f^\alpha\left((1-\alpha)\boldsymbol{q}_h^* + \alpha(\boldsymbol{\Phi}_{hh}\boldsymbol{q}_h^* + \boldsymbol{\Phi}_{ho}\boldsymbol{x}_o + \boldsymbol{b}_h)\right)$. and where the last equality comes from the standard application of the implicit function theorem as typical in DEQs or monotone DEQs. This backward pass can also be computed via an iterative approach, and here the details exactly mirror that of Winston & Kolter (2020).

As a final note, we also mention that owning to the restricted range of weights allowed by the monotonicty constraint, the actual output marginals $q_i(x_i)$ are often more uniform in distribution than desired. Thus, we typically apply the loss to a scaled marginal

$$\tilde{q}_i(x_i) \propto q_i(x_i)^{\tau_i} \tag{20}$$

where $\tau_i \in \mathbb{R}_+$ is a variable-dependent learnable temperature parameter. Importantly, we emphasize that this is *only* done after convergence to the mean-field solution, and thus only applies to the marginals to which we apply a loss: the actual internal iterations of mean-field cannot have such a scaling, as it would violate the monotonicity condition.

## 4 EXPERIMENTAL EVALUATION

We evaluate the performance of our model primarily on the MNIST dataset. Our model is able to approximate any conditional distribution. Here we demonstrate how to model missing pixels and the class digits conditioned on the observed pixels, as well as model missing pixels conditioned on class digits and observed pixels. Although MNIST is of course a small-scale problem, the goal here is to demonstrate joint inference and learning over what is still a reasonably-sized joint model, considering the number of hidden units. Nonetheless, the current experiment is admittedly largely a *demonstration* of the proposed method rather than a full accounting of its performance, a point we highlight in the subsequent section as well. After describing the MNIST experiments, we detail a similar experiment on CIFAR-10 to demonstrate the potential for scaling the approach.

We also show how our inference method differs from the previous ones. On the joint imputation and classification task, we train models using our updates and the updates in Krähenbühl & Koltun (2013); Baqué et al. (2016), and inference each using all three update methods, with and without the monotonicity constraint. We show that our inference method is superior in both performance and convergence speed. Numerically, we also demonstrate that the other two methods could either diverge or not converging to the actual mean-field fixed point. All deferred experiments and details can be found in the appendix.

**Experiment setup** The original MNIST dataset has one channel representing the gray-scale intensity, ranging between 0 and 1. Here we adopt the strategy of Van Oord et al. (2016) to convert this continuous distribution to a discrete one. We bin the intensity evenly to 4 categories $\{0, \ldots, 3\}$, and for each channel uses a one-hot encoding of the category so that the input data has shape $4 \times 28 \times 28$. We remark that the number of categories is chosen arbitrarily and can be any integer. For image with missing data, if the pixels are randomly masked, we always mask each pixel off independently with probability $60\%$, such that in expectation only $40\%$ pixels are observed. If a whole patch of pixels is masked, we randomly pick a $14 \times 14$ patch. The patches/masks are chosenly differently for every image, similar to the query training in Lázaro-Gredilla et al. (2020). To make the model class richer, in the patch case we lift the monotonicity constraint, and the model converges regardless. We also conduct the same set of MNIST experiments using a 3-layer DBM and include the results in the appendix.

**Results on imputation and classification** Training the above described model for 40 epochs with $40\%$ missing pixels leads to a $92.95\%$ test accuracy, and the image reconstruction result is shown in fig. 4d. Based on the reconstructed images and our classification results, our model successfully recovers the unimodal distribution conditioned on the observed pixels. Despite working simultaneously on two different tasks, our model can simply be trained jointly, taking full advantages of the existing auto-differentiation frameworks without additional burden. It is worth noting that while the autoregressive models of Van Oord et al. (2016) are comparable to ours, they require sampling pixels one-by-one in a sequential order at inference time, whereas our model does not assume an underlying sequence and can generate all pixels at once. Comparing to the differently-parameterized monDEQ in Winston & Kolter (2020), whose linear module suffers from drastically increasing condition number (hence in later epochs taking around 20 steps to converge, even with tuned $\alpha$), our

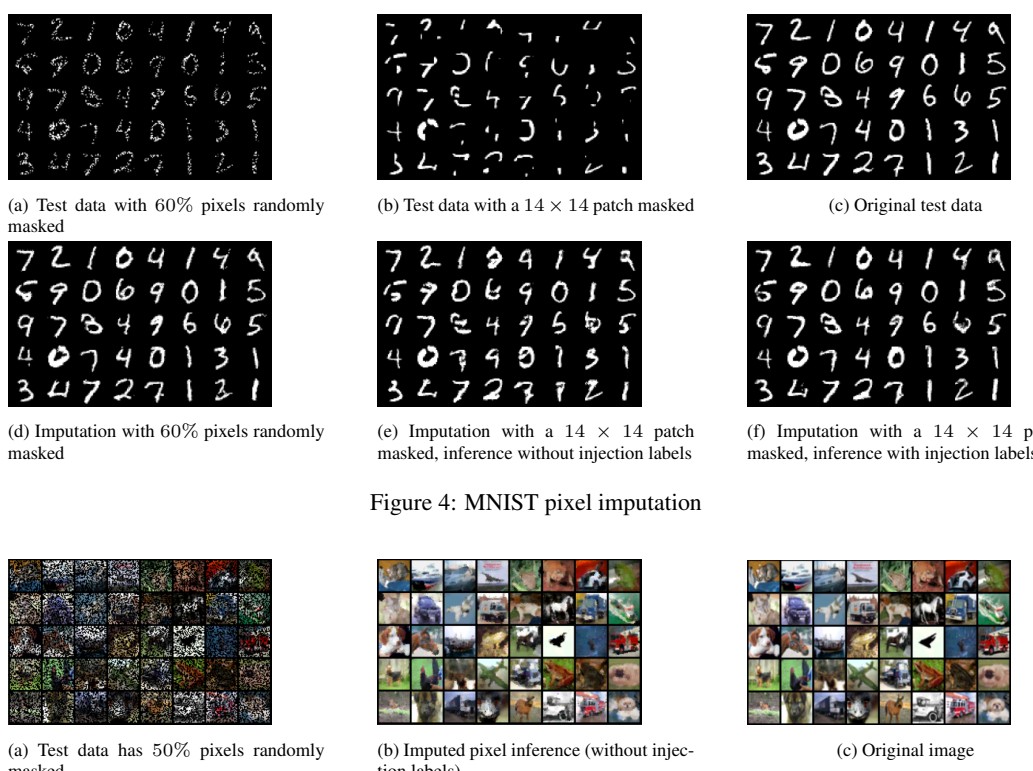

(a) Test data with 60% pixels randomly masked

(b) Test data with a $14 \times 14$ patch masked

(c) Original test data

(d) Imputation with 60% pixels randomly masked

(e) Imputation with a $14 \times 14$ patch masked, inference without injection labels

(f) Imputation with a $14 \times 14$ patch masked, inference with injection labels

Figure 4: MNIST pixel imputation

(a) Test data has 50% pixels randomly masked

(b) Imputed pixel inference (without injection labels)

(c) Original image

Figure 5: CIFAR-10 pixel imputation

parameterization produces a much nicer convergence pattern: the average number of forward iterations over the 40 training epochs is less than 6 steps, see fig. 3. When the missing pixels form consecutive patches, our model reconstructs readable digits despite potentially large chunk of missing pixels (fig. 4e). Meanwhile, if the model is given the image labels as input injections, our model performs conditionaly generation fairly well (fig. 4f). These results demonstrate the flexibility of our parameterization for modelling different conditional distributions.

**CIFAR-10 Experiment**    We additionally conduct an experiment on the simultaneous tasks of image pixel imputation and label prediction given partially observed features. Model architecture and training details are given in the appendix. With 50% of the pixels observed, the model obtains 58% test accuracy, and can impute the missing pixels effectively (see fig. 5).

## 5    CONCLUSION

In this work, we give a monotone parameterization for general Boltzmann machines, and connect its mean-field fixed point to a monotone DEQ model. We provide a mean-field update method that is proven to be globally convergent. Our parameterization allows for full parallelization of mean-field updates without restricting the potential function to be concave, thus addressing issues with prior approaches. Moreover, we allow complicated and hierarchical structures among the variables and show how to efficiently implement them. For parameter learning, we directly optimize the marginal-based loss over the mean-field variational family, circumventing the intractability of computing the partition function. Our model is evaluated on the MNIST and CIFAR-10 dataset for simultaneously predicting with missing data and imputing the missing data itself. There are several potential future directions. First, the monotone parameterization in theorem 3.1 is not an if-and-only-if statement, hence potentially making our model more restrictive than necessary. The second direction is that although we have a fairly efficient implementation of $\mathrm{prox}_f^\alpha$, it is still slower than normal nonlinearities like ReLU or softmax. It is an interesting direction to more efficiently scale these models.

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

## A APPENDIX

### A.1 DEFERRED PROOFS

*Proof of theorem 3.2.* By definition, the proximal operator induced by $f$ (the same $f$ in eq. (11)) and $\alpha$ solves the following optimization problem:

$$\min_z \quad \frac{1}{2}\|x - z\|^2 + \alpha \sum_i z_i \log z_i - \frac{\alpha}{2}\|z\|^2$$
$$\text{s.t.} \quad z_i \geq 0, \ i = 1, \ldots, d,$$
$$\sum_i z_i = 1$$

of which the KKT condition is

$$-x_i + z_i + \alpha + \alpha \log z_i - \alpha z_i + \lambda - \mu_i = 0, \text{for } i \in [d]$$
$$\mu_i \geq 0$$
$$z_i \geq 0$$
$$\sum_{i \in [d]} \mu_i z_i = 0$$
$$\sum_{i=1}^d z_i = 1$$

We have that $\mu_i = 0$ is feasible and the first equation of the above KKT condition can be massaged as

$$-x_i + z_i + \alpha + \alpha \log z_i - \alpha z_i + \lambda - \mu_i = 0$$
$$\Longleftrightarrow (1 - \alpha)z_i + \alpha \log z_i = x_i - \alpha - \lambda$$
$$\Longleftrightarrow \frac{(1 - \alpha)z_i + \alpha \log z_i}{\alpha} = \frac{x_i - \alpha - \lambda}{\alpha}$$
$$\Longleftrightarrow \exp\left(\frac{(1 - \alpha)z_i + \alpha \log z_i}{\alpha}\right) = \exp\left(\frac{x_i - \alpha - \lambda}{\alpha}\right)$$
$$\Longleftrightarrow z_i \exp\left(\frac{1 - \alpha}{\alpha} z_i\right) = \exp\left(\frac{x_i - \alpha - \lambda}{\alpha}\right)$$
$$\Longleftrightarrow \frac{1 - \alpha}{\alpha} z_i \exp\left(\frac{1 - \alpha}{\alpha} z_i\right) = \frac{1 - \alpha}{\alpha} \exp\left(\frac{x_i - \alpha - \lambda}{\alpha}\right)$$
$$\Longleftrightarrow \frac{1 - \alpha}{\alpha} z_i = W\left(\frac{1 - \alpha}{\alpha} \exp\left(\frac{x_i - \alpha - \lambda}{\alpha}\right)\right)$$

where $W$ is the lambert W function. Notice here $z_i > 0$. Our primal problem is convex and Slater's condition holds. Hence, we conclude that

$$z_i = \frac{\alpha}{1 - \alpha} W\left(\frac{1 - \alpha}{\alpha} \exp\left(\frac{x_i - \alpha - \lambda}{\alpha}\right)\right).$$

$\square$

### A.2 CONVOLUTION NETWORK

It is clear that the monotone parameterization in section 3 directly applies to fully-connected networks, and all the related quantities can be calculated easily. Nonetheless, the real power of the DEQ model comes in when we use more sophisticated linear operators like convolutions. In the context of Boltzmann machines, the convolution operator gives edge potentials beneficial structures. For example, when modeling the joint probability of pixels in an image, it is intuitive that only the nearby pixels depend closely on each other.

Let $A \in \mathbb{R}^{k \times k \times r \times r}$ denote a convolutional tensor with kernel size $r$ and channel size $k$, let $x$ denote some input. For a convolution with stride 1, the block diagonal elements of $A^T A$ simply form a $1 \times 1$ convolution. In particular, we apply the convolutions

$$- A^T(A(x)) + \tilde{A}(x) \tag{21}$$

where $\tilde{A}$ is a $1 \times 1$ convolution given by

$$\tilde{A}[:,:] = \sum_{i,j} A[:,:,i,j]^T A[:,:,i,j]. \tag{22}$$

We can normalize by the spectral norm of $\tilde{A}$ term to ensure strong monotonicity. Since $\tilde{A}$ can be rewritten as a $k \times k$ matrix and $k$ is usually small, its spectral norm can be easily calculated.

It takes more effort to work out convolutions with stride other than 1. Specifically, the block diagonal terms do not form a $1 \times 1$ convolution anymore, instead, the computation varies depending on the location. It is easier to see the computation directly in the accompanying code.

**Grouped channels**    It is crucial to introduce the concept of grouped channels, which allows us to represent multiple categorical variables in a single location, such as the three categorical variables representing the three (binned) color channels of an RGB pixel. In this case, each of the three RGB channels will be represented by a different group of $k$ channels representing the $k$ bins. The grouping is achieved by having the nonlinearity function (softmax) applied to each group separately. We remark that the convolutions themselves are *not* grouped, otherwise none of the red pixels would interact with green or blue pixels, etc. Instead, we want all RGB channels to interact with each other (except that channel $i$ at position $(j, k)$ does not interact with itself). That means in eq. (4), the $\mathrm{blkdiag}(\hat{A}^T \hat{A})$ is grouped in the following way. Recall that this block diagonal term has element of size $k_i \times k_i$ for $i \in [n]$. This parameterization has only 1 group. With $g$ groups, the element of the block diagonal matrix then has size $k_{i_1} \times k_{i_1}, \ldots, k_{i_g} \times k_{i_g}$ for $i \in [n]$, where $\sum_{j \in [g]} k_{i_j} = k_i$. We also observe empirically that grouping the latent variables improves the performance.

## A.3    EFFICIENT COMPUTATION OF $\mathrm{prox}_f^\alpha$

The solution to the proximal operator in damped forward iteration given in theorem 3.2 involves the Lambert W function, which does not attain an analytical solution. In this section, we show how to efficiently calculate the nonlinearity $\sigma(x_i)$, as well as its Jacobian matrix for backward iteration.

Let $f(y) = \frac{\alpha}{1-\alpha} W\left(\frac{1-\alpha}{\alpha} \exp\left(\frac{y}{\alpha} - 1\right)\right)$, and we have

$$x = \log f(y) = \log \frac{\alpha}{1-\alpha} + \log e^{y/\alpha - 1} + \log \frac{1-\alpha}{\alpha} - W\left(\frac{1-\alpha}{\alpha} \exp\left(\frac{y}{\alpha} - 1\right)\right),$$

where the last equality uses the identity $\log(W(x)) = \log x - W(x)$. Rewrite $W\left(\frac{1-\alpha}{\alpha} \exp\left(\frac{y}{\alpha} - 1\right)\right) = f(y)\frac{1-\alpha}{\alpha}$ and massage the terms, we have that solving $\log f(y)$ is equivalent to finding the root of

$$h(x) = y - \alpha - e^x(1-\alpha) - \alpha x.$$

Direct calculation shows that $h'(x) = -\alpha - (1-\alpha)e^x$ and $h''(x) = -(1-\alpha)e^x$. Note here $y$ is the input and it is known to us, and $x$ is a scalar. Hence we can efficiently solve the root finding problem using Halley's method. For backpropagation, we need $\frac{dx}{dy}$, which can be computed by implicit differentiation:

$$h(x) = y - \alpha - e^x(1-\alpha) - \alpha x = 0$$

$$\implies \frac{dx}{dy} = \frac{1}{\alpha + (1-\alpha)e^x} = \frac{1}{y - \alpha x}.$$

Now we can find $\lambda$ s.t $\sum_i z_i = 1$ using Newton's method on $g(\lambda) = \sum_i e^{\log(f(x_i + \lambda))} - 1 = 0$. Note this is still a one-dimensional optimization problem. A direct calculation shows that $\frac{dg}{d\lambda} = \sum_i e^{\log(f(x_i + \lambda))} \frac{d \log(f(x_i + \lambda))}{d\lambda}$, and above we have already calculated that

$$\frac{d \log(f(x_i + \lambda))}{d\lambda} = \frac{dx^*}{dy} = \frac{1}{y + \lambda - \alpha x}.$$

For backward computation, by the chain rule, we have:

$$
\frac{de^{\log f(x_i+\lambda)}}{dx_i} = e^{\log f(x_i+\lambda)} \frac{d\log(f(x_i+\lambda))}{dx_i}
$$

$$
= e^{\log f(x_i+\lambda)} \frac{1 + d\lambda/dx_i}{x_i + \lambda - \alpha \log(f(x_i+\lambda))},
$$

where the last step is derived by implicit differentiation. Now to get $d\lambda/dx_i$, notice that by applying the implicit function theorem on $p(x, \lambda(x)) = \sum_i e^{\log(f(x_i+\lambda))} - 1 = 0$, we get

$$
\frac{d\lambda}{dx_i} = -\left(\frac{dp}{d\lambda}\right)^{-1} \frac{dp}{dx_i}.
$$

Thus we have all the terms computed, which finishes the derivation.

## B  ADDITIONAL EXPERIMENTS AND DETAILS

Here we provide the model architectures and experiment details omitted in the main text.

**Model architecture**    For MNIST experiments, using the notation in eq. (13), we design a 20-layer deep monotone DEQ with the following structure:

$$
\begin{bmatrix}
\boldsymbol{A}_{11} & 0 & 0 & 0 \\
\boldsymbol{A}_{21} & \boldsymbol{A}_{22} & 0 & 0 \\
\boldsymbol{A}_{31} & \boldsymbol{A}_{32} & \boldsymbol{A}_{33} & 0 \\
0 & 0 & \boldsymbol{A}_{43} & \boldsymbol{A}_{44}
\end{bmatrix},
$$

where $\boldsymbol{A}_{11}$ is a $20\times20\times3\times3$ convolution, $\boldsymbol{A}_{22}$ is a $40\times40\times3\times3$ convolution, $\boldsymbol{A}_{21}$ is a $40\times20\times3\times3$ convolution with stride 2, $\boldsymbol{A}_{33}$ is a $80\times80\times3\times3$ convolution, $\boldsymbol{A}_{31}$ is a $80\times20\times3\times3$ convolution with stride 4, $\boldsymbol{A}_{32}$ is a $80\times40\times3\times3$ convolution with stride 2, $\boldsymbol{A}_{43}$ is a $(80\cdot7\cdot7)\times10$ dense linear layer, and $\boldsymbol{A}_{44}$ is a $10\times10$ dense linear layer. The corresponding variable $\boldsymbol{q}$ as in eq. (12) then has 4 elements of shape $(20\times28\times28), (40\times14\times14), (80\times7\times7), (10\times1)$. When applying the proximal operator to $\boldsymbol{q}$, we use $1, 10, 20, 1$ as their number of groups, respectively.

**Training details and hyperparameters**    Treating the image reconstruction as a dense classification task, we use cross-entropy loss and class weights $\frac{1-\beta}{1-\beta^{n_i}}$ with $\beta = 0.9999$ (Cui et al., 2019), where $n_i$ is the number of times pixels with intensity $i$ appear in the hidden pixels. For classification, we use standard cross-entropy loss. To enable joint training, we put equal weight of $0.5$ on both task losses and backpropagate through their sum. For both tasks, we put $\tau_i \boldsymbol{\Phi} \boldsymbol{q}_i^*$ into the cross-entropy loss as logits, as described in eq. (20). Since mean-field approximation is (conditionally) unimodal, the scaling grants us the ability to model more extreme distributions. To achieve faster damped forward-backward iteration, we implement Anderson acceleration (Walker & Ni, 2011), and stop the fixed point update as soon as the relative difference between two iterations (that is, $\|\boldsymbol{q}_{t+1} - \boldsymbol{q}_t\|/\|\boldsymbol{q}_t\|$) is less than $0.01$, unless we hit a maximum number of 50 allowed iterations. For $\text{prox}_f^\alpha$ and the damped iteration, we set $\alpha = 0.125$ (Although one can tune down $\alpha$ whenever the iterations do not converge, empirically this never happens on our task). We use the Adam optimizer with learning rate $0.001$. Our models are trained on one GeForce GTX 2080 Ti GPU.

**Comparison to past inference methods**    We conduct numerical experiments to compare our inference updating method to the ones proposed by Krähenbühl & Koltun (2013); Baqué et al. (2016), denoted as Krähenbühl's and Baqué's respectively. Krähenbühl's fast concave-convex procedure (CCCP) essentially decomposes to eq. (14), the un-damped mean-field update with softmax. This update only converges provably when $\boldsymbol{\Phi}$ is concave. Baqué's inference method can be written as

$$
\boldsymbol{q}_h^{(t)} = \text{softmax}\left((1-\alpha)\log\boldsymbol{q}_h^{(t-1)} + \alpha(\boldsymbol{\Phi}_{hh}\boldsymbol{q}_h^{(t-1)} + \boldsymbol{\Phi}_{ho}\boldsymbol{x}_o + \boldsymbol{b}_h)\right). \tag{23}
$$

This algorithm provably converges despite the property of the pairwise kernel function. However, this procedure converges in the sense that the variational free energy keeps decreasing. Therefore their fixed point may not be the true mean-field distribution eq. (10). In this experiment, we train the

Table 1: Relative update residual when monotonicity is enforced

| Inference / Train | Krähenbühl | Baqué | Our |
|---|---|---|---|
| Krähenbühl | 0.0004 | 0.0061 | 0.0024 |
| Baqué | 1.250 | 0.0059 | 0.0024 |
| Our | 1.144 | 0.0057 | 0.0017 |

Table 2: Relative update residual when monotonicity is not enforced

| Inference / Train | Krähenbühl | Baqué | Our |
|---|---|---|---|
| Krähenbühl | 0.0005 | 0.0065 | 0.0024 |
| Baqué | 1.0924 | 0.0119 | 0.0042 |
| Our | 1.1286 | 0.0065 | 0.0022 |

models using three different updating methods, and perform inference using three methods as well, with and without the monotonicity condition.

Krähenbühl's and Baqué's methods often do not converge in the backward pass (there's no theoretical guarantees neither). To rule out the impact of the backward iteration, during training we directly update use the gradient of the forward pass, instead of using a backward gradient hook to compute eq. (19). Figure 7 and fig. 8 demonstrate how the three update methods impute missing pixels when trained with different update rules, with and without the monotonicity condition, respectively. Krähenbühl's usually does not converge when the model is trained with our method or Baqué's, whereas the other two methods impute the missing pixels well. The classification results are presented in table 3 and table 4. Notice that when trained with our method or Baqué's, the convergence issue of Krähenbühl's leads to horrible classification accuracy. Our method is superior to other inference methods when the model is trained in a different update fashion. For example, if the model is trained by using Krähenbühl's, it makes sense that the model performs the best if the inference is also Krähenbühl's since the parameters are biased toward that particular inference method. However, our method in this case outperforms Baqué's.

After these methods halt and return $\boldsymbol{q}_h^T$, we run one more iteration of

$$\boldsymbol{q}_h^{T+1} = \text{softmax}\left(\boldsymbol{\Phi}_{hh}\boldsymbol{q}_h^T + \boldsymbol{\Phi}_{ho}\mathbf{x}_o + \boldsymbol{b}_h\right), \tag{24}$$

and record the relative update residual $\|\boldsymbol{q}_h^{T+1} - \boldsymbol{q}_h^T\|/\|\boldsymbol{q}_h^T\|$ for randomly selected 4000 MNIST images. The results are listed in table 1 and table 2. To alleviate the effect of numerical issues, we strength the convergence condition to either the relative residual is less than $10^{-3}$ or the number of iterations exceeds 100 steps.

It appears in table 1 and table 2 that although our method has a much lower residual compare to Baqué's, both of them seem small and convergent. This is because the "optimal" fixed point in this setting on MNIST might be unique and both methods happen to converge to the same point. However, this is in general not true. We compare our method vs Baqué's on 400 randomly selected MNIST test images with 40% pixels observed, and perform mean-field update until the relative residual of $[0.1, 0.05, 0.01, 0.005, 0.001, 0.0005, 0.0001]$ is reached (without step constraint), respectively. Then we measure the TV distance between the distributions computed by these two methods on the remaining 60% pixels, as well as the convergence speed. The results are demonstrated in fig. 6. One can see that when the model is trained (using Krähenbühl's, fig. 9a), the TV distance converges to 0 as the tolerance decreases. However, when the model is just initialized (but still constrained to be monotone), the TV distance remains large (fig. 6c). Even though in this case the optimal fixed point may be unique, our method is still superior to Baqué's: it takes us less iterations till convergence, despite whether the model is trained or not..

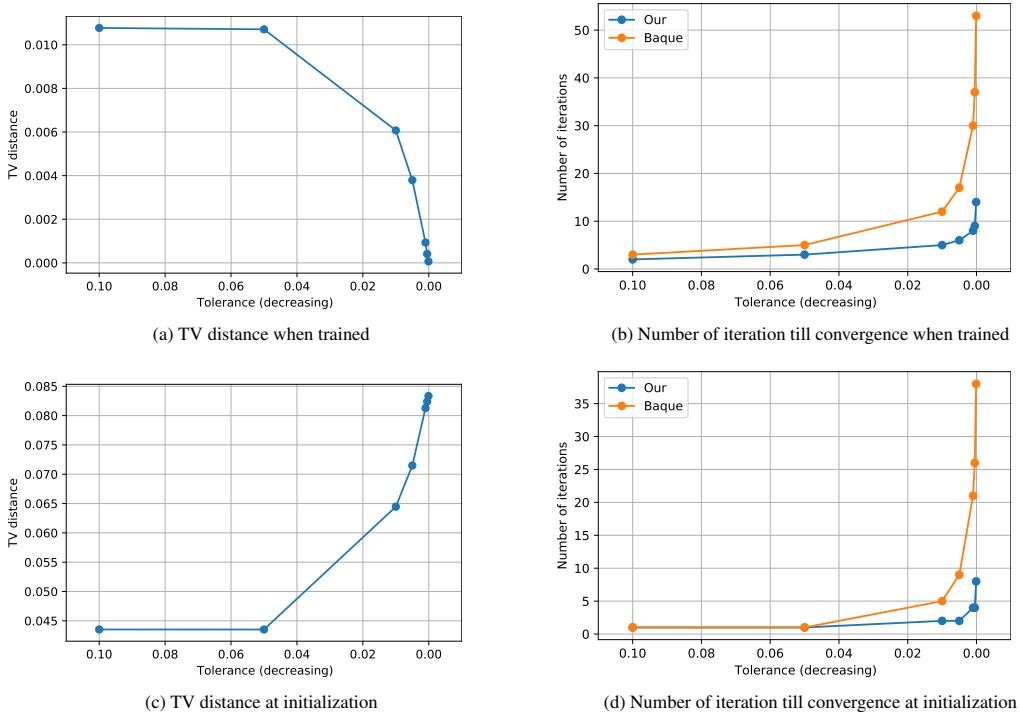

(a) TV distance when trained

(b) Number of iteration till convergence when trained

(c) TV distance at initialization

(d) Number of iteration till convergence at initialization

Figure 6: TV distance and convergence speed

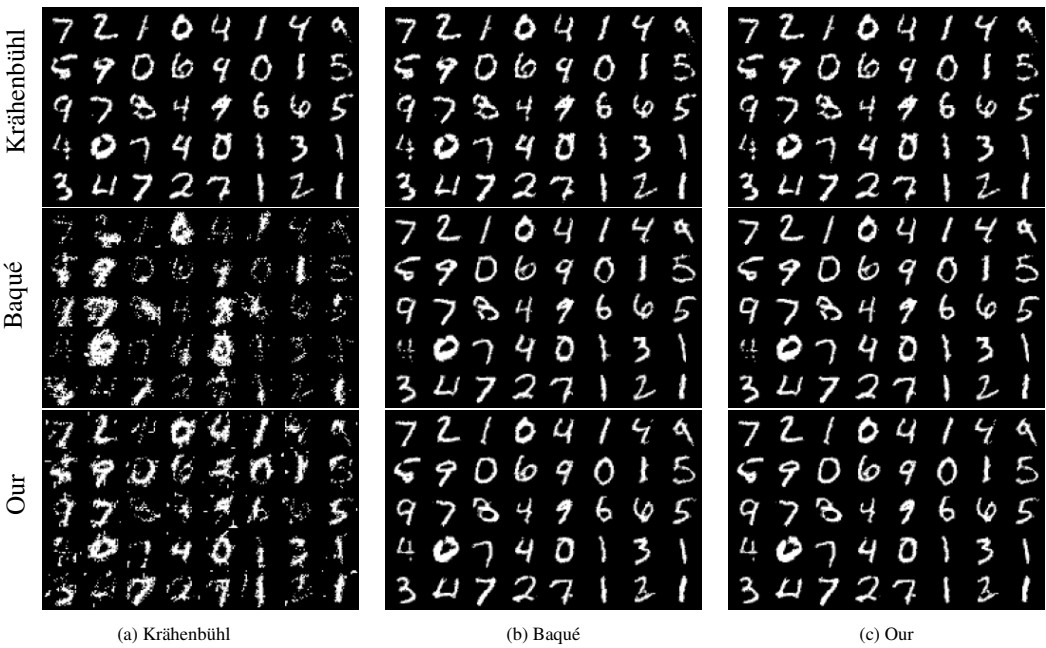

(a) Krähenbühl

(b) Baqué

(c) Our

Figure 7: Training and inference using all three update rules with $40\%$ observed pixels *with* the monotonicity condition. The labels on each row represent the training update rule, and the labels on the columns represent the inference update rule.

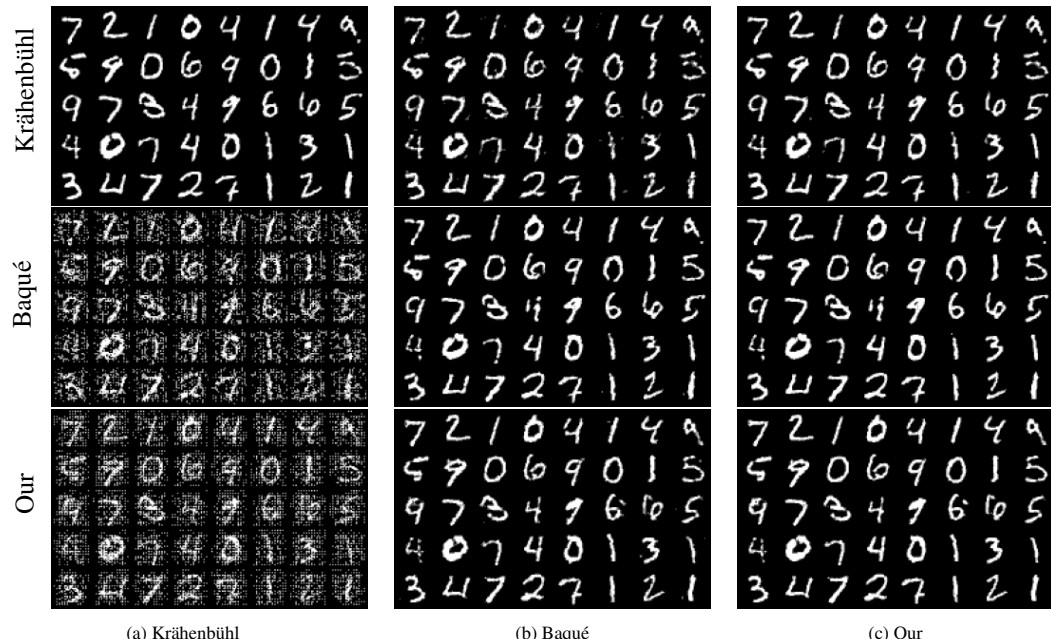

|          | (a) Krähenbühl | (b) Baqué | (c) Our |
|----------|----------------|-----------|---------|

Figure 8: Training and inference using all three update rules with $40\%$ observed pixels *without* the monotonicity condition. The labels on each row represent the training update rule, and the labels on the columns represent the inference update rule.

Table 3: Classification error (standard deviation) when monotonicity is enforced

| Inference / Train | Krähenbühl | Baqué | Our |
|---|---|---|---|
| Krähenbühl | **0.042 (0.0013)** | 0.114 (0.0019) | 0.0498 (0.0014) |
| Baqué | 0.958 (0.0013) | 0.038 (0.0010) | **0.034 (0.0012)** |
| Our | 0.946 (0.0024) | 0.0425 (0.0016) | **0.0412 (0.0017)** |

**CIFAR-10 model architecture** Architecture is the same as for the MNIST experiments with the following exceptions: $A_{11}$ is a $20 \times 20 \times 3 \times 3$ convolution, $A_{22}$ is a $24 \times 24 \times 3 \times 3$ convolution, $A_{21}$ is a $24 \times 20 \times 3 \times 3$ convolution with stride 2, $A_{33}$ is a $48 \times 48 \times 3 \times 3$ convolution, $A_{31}$ is a $48 \times 20 \times 3 \times 3$ convolution with stride 4, $A_{32}$ is a $48 \times 24 \times 3 \times 3$ convolution with stride 2, $A_{43}$ is a $(48 \cdot 8 \cdot 8) \times 10$ dense linear layer, and $A_{44}$ is a $10 \times 10$ dense linear layer. The corresponding variable $q$ as in eq. (12) then has 4 elements of shape $(60 \times 32 \times 32)$, $(24 \times 16 \times 16)$, $(48 \times 8 \times 8)$, $(10 \times 1)$. When applying the proximal operator to $q$, we use $1, 6, 12, 1$ as their number of groups, respectively.

**CIFAR-10 training details** Training details are the same as for the MNIST experiments with the following exceptions: The model is trained for 100 epochs using standard data augmentation. During the first 10 epochs, the weight on the reconstruction loss is ramped up from 0.0 to 0.5 and the weight on the classification loss ramped down from 1.0 to 0.5. Also during the first 20 epochs, the percentage of observation pixels is ramped down from $100\%$ to $50\%$.

## C  COMPARISON TO RBM/DBM

We conduct the same set of experiments on MNIST using DBM for comparison. We use a 3-layer DBM where the first hidden layer has 300 neurons, and the last hidden layer (representing the digits) has 10 neurons, amounting to in total 238,200 parameters. Our proposed model in the MNIST experiments uses 165,300 parameters.

For image imputation, we randomly mask off $60\%$ pixels, or a randomly selected $14 \times 14$ patch; the results are shown in fig. 9. In the experiment with $60\%$ of pixels randomly masked, we also test

Table 4: Classification error (standard deviation) when monotonicity is not enforced

| Inference / Train | Krähenbühl | Baqué | Our |
|---|---|---|---|
| Krähenbühl | **0.035 (0.0017)** | 0.189 (0.0023) | 0.051 (0.0015) |
| Baqué | 0.762 (0.0013) | **0.041 (0.0013)** | 0.055 (0.0012) |
| Our | 0.90 (0.0002) | 0.063 (0.0021) | **0.036 (0.0017)** |



(a) 60% pixels are randomly masked. From left to right: imputed image, true image, masked image.



(b) 14 × 14 patches are randomly masked. From left to right: imputed image, true image, masked image.

Figure 9: DBM for image imputation

the model on predicting the actual digit simultaneously. The test accuracy is 93.58%. Our model achieves comparable test accuracy (92.95%) and imputation compared to this DBM, given fewer parameters and despite the the monotonicity constraint of our model.

The DBM is trained using $CD$-1 algorithm for 100 epochs with a batch size of 128 and learning rate of 0.01. For imputation and classification, the DBM uses Gibbs sampling of 100 steps, although the quality of the imputed image and test accuracy are insensitive to the number of steps.

