# OpenReview forum: "Monotone deep Boltzmann machines"
_ICLR.cc/2022/Conference — ICLR 2022 Submitted_

### Official Review · Reviewer_fJmJ · 2021-10-30

**Correctness:** 4
**Technical Novelty And Significance:** 2
**Empirical Novelty And Significance:** 2
**Recommendation:** 5
**Confidence:** 4

**Main Review:**

The paper is very well-written and easy to read. However, I found the novelty aspect of the work to be a bit lacking:
- Aside from the new parameterization Eq (3),(4) introduced to satisfy the monotonicity condition, the method of this paper seems like a straightforward combination of [Krahenbuhl & Kolten 2013], [Baque et al. 2016], and [Winston & Kolter 2020].
- It is also unclear how restirctive this parameterization is (which itself is quite simple) within all possible pairwise potentials that satisfy the monotonicity condition.
- The parallel updates and the convergence proof are almost exactly the same as [Winston & Kolter 2020], except for the extension to softmax operation.

I would be happy with the novelty aspect if convincing experiments results are shown. Sadly this does not seem to be the case:
- The practical benefit of deep Boltzmann machine compared to more traditional neural architectures (e.g. CNN for image classification) is not clear to me and has not been highlighted in the paper. When would someone use deep BM instead of the alternatives? The experimental results do not seem to answer this question.
- Although deep Boltzmann machine can be more flexible for modeling different conditional distributions without retraining, it seems to come at the cost of being much harder to train while relying on mean-field approximation. I'm wondering how crude the mean-field approximation of the posterior distribution is in the current paper's setting, which has not been discussed.
- The experiments are very small-scaled. The images are all with very low-resolution. This seems to suggest the impracticality of deep Boltzmann machine. In CIFAR-10 experiment, the test accurarcy is only 58%, which is a lot lower than using conventional neural architectures.
- In Eq (20) a very arbitrary scaling is used after convergence to the mean-field solution. This seems like an ad-hoc fix for a method that doesn't really work due to the monotonicity constraint. I'd be interested in seeing experimental comparison between the scaled version and the original version.
- For the patch case, the model works better without the monotonicity constraint. This seems to be against the whole point of the paper.
- The proposed method does not seem to have significant improvement compared to past works in this line of work (e.g. diagonal entries in Table 3, 4).

Additional comments:
- Page 3, "We remark the readers upon ..." this doesn't sound grammatically correct.
- Sec 3.5 mentions the model is trained directly to output correct marginals, instead of the usual likelihood maximization, which can be intractable. What is lost in this simplification (in addition to mean-field approximation)? Matching only marginals seems very coarse to me.
- How to train the proposed model on batches of images? If I understand correctly, the current training procedure would sample a single image, split it into $x_h$ and $x_o$, run mean-field inference given $x_o$ in a differentiable manner, then backpropagate through loss $\ell(q_h^*, x_h)$. Are multiple mean-field inferences run in parallel? If so do they use the same number of iterations? If not, I would imagine the training to have very high variance.
- At the end of Sec 3.5, at the top of page 8, why is $g(q_h^*)$ not the damped version?


**Summary Of The Paper:**

This paper proposes a new family of monotone deep Boltzmann machines where the pairwise potentials satisfy a monotonicity condition, giving rise to efficient mean-field iteration with provable convergence guarantees. The convergence is obtained by drawing connections with monotone deep equilibrium models. Small-scale experiments are done as proof of concept.


**Summary Of The Review:**

This paper is well-written but its contributions are incremental with somewhat weak experimental results.

---

> ### Author Response · Authors · 2021-11-14
> **Thanks for your review**
>
> We appreciate the valuable comments by the reviewer. We respectfully disagree that this work is a straightforward combination of previous works. As we mentioned in the paper, the proximal function in eq(15) and the monotonicity condition on the hollow matrix $\Phi$ in eq(5) are very non-trivial to deal with in practice, especially when the network topology is complicated (e.g. including strided convoltuion). While the convergence proof follows from Winston & Kolter, our parameterization allows a probabilistic interpretation (thanks to the proximal function and the hollowness of $\Phi$), while their work does not.
>
> Since the model we propose is a probabilistic model, it is not appropriate to compare to, say, CNNs for classification. Joint probabilistic models like RBM (and ours) can model any conditional distribution, while a vanilla CNN cannot. For example, an RBM can perform imputation and classification simultaneously, but a traditional CNN may not. And, while it is true that our experiments are small-scale compared to the tasks to which CNNs can be applied, they are still on a reasonable scale compared to past work on probabilistic models.
>
> Additional comments:
>
> 1. Our task here only considers to model conditional distribution. What is lost using the marginal-based loss is the ability to model joint distribution.
> 2. Running our mean-field update in parallel is exactly the same as running a DEQ model: you can sample a batch of images, and update all of them simultaneously. The stopping criterion is users' choice: you can specify a threshold at which you stop the training. In our experiment, we stop the update whenever the relative residual is less than 0.01, where the residual is the norm of the difference between two iterations on all the samples (more specifically, $\|q^t-q^{t-1}\|/(q^{t-1})$. If $q^t$ represents a distribution over image pixels, then it's of shape (batch_size, channel, width, height)).
> 3. We have a typo in defining $g$, thanks for pointing it out.

---

> > ### Comment · Reviewer_fJmJ · 2021-11-20
> > **Thank you**
> >
> > Thank you for the reply. I remain unconvinced about the practicality of the deep Boltzmann machine demonstrated in this work, although the same challenges (e.g. conditional inference) exist for most probablistic graphical models. Many points in my original review remain unaddressed, such as the restrictiveness of the parameterization, the arbitrary scaling in Eq (20), etc. For these reasons, I decided to keep my score unchanged.

---

> > > ### Author Response · Authors · 2021-11-23
> > > **Thanks for the update**
> > >
> > > Thank you for your update. This may not change your opinion, but we also include a comparison to RBM/DBM and submitted a revision. We hope the revision can be a completion to our previous response.

---

### Official Review · Reviewer_ihMA · 2021-10-31

**Correctness:** 4
**Technical Novelty And Significance:** 4
**Empirical Novelty And Significance:** 1
**Recommendation:** 3
**Confidence:** 4

**Main Review:**

## Strength

The strength of this paper is the technical contribution of finding the connection between the DBMs and the DEQ model by characterizing the monotone DBMs. I think this is a good contribution as it can be essential for further development of BMs, considering that the current progress of BMs is not so rapid, in my understanding.

The quality of presentation is also good, and this paper is clearly written overall. I have just a minor comment:
- Since the current explanation of a block hollow matrix is vague, please mathematically define it for the self-completeness.

## Weaknesses

The significance of this paper is not high, and evaluation is weak. In particular, the practical advantage of the proposed monotone DBMs is not clear.

Although it is true that the family of BMs to which the contrastive divergence algorithm can be applied is limited, any BMs with any connection patterns of hidden variables can be trained by directly applying Gibbs sampling, which of course includes the monotone DBMs.
Therefore, the monotone DBMs have no merit with respect to the effectiveness of inference, and I guess the only practical advantage of the monotone DBMs can be the efficiency. However, there is neither analysis of computational complexity nor empirical runtime comparison to such a straightforward approach.

In addition, it has been already proven that RBMs can represent any distribution, therefore, from the viewpoint of the representation power, there is no difference between RBMs and monotone DBMs. Of course, I agree that monotone DBMs can be more effective than RBMs and existing DBMs in practice, for example, monotone DBMs can achieve more accurate inference with less parameters than RBMs. However, there are no such comparisons in this paper.

I am happy to increase my score if the above my concerns are properly addressed by the authors' response.

**Summary Of The Paper:**

This paper theoretically shows that the mean-field equation for a certain family of Boltzmann machines with hidden variables, called the monotone DBMs, can be modeled as the recently proposed monotone Deep Equilibrium (DEQ) model. This paper further characterizes properties of such Boltzmann machines and its training, and shows its behavior in experiments on MNIST and CIFAR-10.


**Summary Of The Review:**

This paper potentially includes an interesting technical contribution, while the significance is not convincing and the evaluation is weak.

---

> ### Author Response · Authors · 2021-11-14
> **Thanks for your review**
>
> We appreciate the valuable comments by the reviewer. Due to the network topology (especially the intra-layer connections), it is computationally infeasible to apply Gibbs sampling in our model. Unlike (deep) RBM, we cannot perform a block Gibbs sampling. In terms of computational complexity, as long as the damping coefficient satisfies $\alpha<2m/L^2$, where $m$ is the monotonicity factor and $L$ is the Lipschitz constant of the linear operator, our parallel mean-field update converges linearly. We will add more discussions in the paper.
>
> Regarding comparison of representation power to that of RBMs, we are working on experiments to this effect, which has been requested by several reviewers.

---

> > ### Comment · Reviewer_ihMA · 2021-11-23
> > **Thank you**
> >
> > Thank you for your reply. I have checked your reply and the revised paper. Unfortunately, I feel that my concerns are not properly addressed in the revision.
> >
> > It is still not clear how efficient the proposed method is compared to RBM/DBM as there is no runtime comparison.
> > Although comparison with DBM is newly added in the revision, which I acknowledge, this experiment is not sufficient. At least comparison with DBMs with the same number of parameters should be performed for fair comparison.

---

### Official Review · Reviewer_8YQo · 2021-11-02

**Correctness:** 4
**Technical Novelty And Significance:** 3
**Empirical Novelty And Significance:** 2
**Recommendation:** 5
**Confidence:** 5

**Main Review:**

The paper is well written and easy to follow. Most of its contents come from existing literature, but this work nicely puts those existing pieces (single fixed point, parallel updates) together, providing a probabilistic interpretation as a Boltzmann machine that is new.

While this paper emphasizes how the proposed approach enables the use of general Boltzmann machines (BM) and not just stacked restricted BMs, the resulting model might actually be more restricted than the stacked RBMs that it intends to improve upon. It is true that the proposed model can contain intra-layer and skip-layer connections that a DBN lacks, but all the parameters are restricted so as to produce a monomodal posterior approximation for _any_ partial evidence. The true posterior, even for a single-layer RBM, can be multimodal if the parameters are not restricted. This means that, as a modeling tool, the proposed BM with restricted weights might be less flexible than a DBN. Many densities of interest are multimodal, particularly as we reduce the available evidence. In fact, in the absence of evidence, any useful BM will have to be multimodal (for instance, to be able to sample different MNIST digits from it).

The proposed mechanism for training is also lacking in that it only allows for marginal supervised learning: it cannot be used for unsupervised learning, which is the typical mode of operation for RBMs and DBNs. Basically, the tasks that it can solve need to be crafted in such a way that the evidence provided is enough to disambiguate a single mode of the posterior. For instance, if we want to perform MNIST digits inpainting and we provide only the top 25% of the image showing a semicircle ⌒, possible completions could be 0, 2, 3, 6 8, 9. This method would fail at this task since it would consistently default to a single digit, or even worse, a single combination of the possible completions.

Thus, the presented paper does provide an efficient mechanism for conditional training of parameter-restricted BMs (and does a good job at it), but the use cases in which it can be applied are severely limited, both due to the type of training and parameters it can use.

The experimental section does not contain meaningful comparisons with other methods/baselines:
- Baseline 1: Use your loss function with damped parallel mean field inference (i.e. consider damping a hyperparameter and do not impose any restriction on the parameters of the BM).
- Baseline 2: Use a DBN (less raw expressive power, but unrestricted in parameters and with a more proper loss function).

So it is difficult to gauge the practical advantage in the provided examples.

Minor comments and questions:
- You show that the mean field inference problem has a single global optimum. But is the true posterior monomodal under this parameterization? That would be a stronger result and convenient to know.

- Is the query (the split between observed variables and variables one wants to predict) fixed throughout training? Although this is not explicitly pointed out in the theoretical part of your paper (it seems to be fixed, citing Domke 2013), this split could be different for each training sample, which seems to be the case based on your experiments. Using different splits is called "query training" in this AAAI 2021 paper "Query Training: Learning a Worse Model to Infer Better Marginals in Undirected Graphical Models with Hidden Variables", which seems to propose a very similar approach, although using a different type of inference. It'd be good to clarify which approach you are using in the description of training.

- The definition of the function in Eq. (11) is a bit confusing because of how the domain is included. Could you define it by parts, or define I(.)?

- Which is the value of alpha that you use for your experiments?

- The figure "92.95% test accuracy" corresponds to the 10-way labels of each digit? Or to the 4-way categories of the pixels?

- Typo: "that owning to the restricted" -> owing

**Summary Of The Paper:**

In this paper the authors propose a restricted parameterization of the Boltzmann machine that guarantees that for any set of observations, the mean field objective has a single global optimum. Furthermore, that global optimum can be provably achieved using damped parallel mean-field updates, which make inference efficient. To turn inference into learning, the model is treated as a supervised learning model: some of its variables are considered to be observed inputs and some of its variables are considered to be target outputs (known at test time). The usual, marginal cross-entropy loss is the optimization target for learning.

**Summary Of The Review:**

Pros: This paper does a good job at providing a mechanism for inference in (parameter restricted) BMs with convergence guarantees, as well as an efficient method to learn the parameters of these BMs.

Cons: The proposed method cannot be applied in many settings in which BMs can (unsupervised learning, sampling, use of multimodal posteriors). Little experimental validation of the usefulness of the convergent inference.

So the settings in which this approach can be used is very limited, but within that setting, it provides the required details for efficient training and robust guarantees for inference.

---

> ### Author Response · Authors · 2021-11-14
> **Thanks for your review**
>
> We appreciate the valuable comments by the reviewer. We agree that the distribution we model is conditional unimodal. By more general than RBM, we mean the network topology is more general. We still think it is interesting to consider conditional unimodal distributions, at least in some cases. The example you gave about image imputation is very accurate, just given the top $25\%$ pixels, our model may not able to correctly impute the rest image. However, it will correctly impute the image if the true digit is also given. This is demonstrated in our experiment, see figure 4 (e)(f). The digit ``3'' in the (2,2) position (index counting from 0, from the top left corner) was not fully imputed without label injection (e), but can be imputed with label injection (f). If more information is given, there's a bigger chance that the true underlying conditional distribution is unimodal. This is a trade-off one must suffer, if a global convergence to a global minimum is guaranteed, then the distribution cannot be multimodal. We believe this convergence guarantee is indeed desired in some cases.
>
> Our current experiment includes your Baseline 1. The results were included in Figure 8 and Table 4 in the appendix. Inference without the monotonicity constraint gives slightly better performance on the MNIST dataset, but usually takes more steps to converge. Regarding baseline 2, we are working on an experimental comparison to RBMs, which has been requested by several reviewers.
>
> Minor comments:
> 1. If by true posterior, you mean $P(h|v)$, where $h$ is the set of hidden variables, $v$ is the set of visible variables, and $P$ is the data distribution, then it is very unlikely to be unimodal.
> 2. The split is not fixed during training (for each training batch, we sample a different set of visible variables), this is to ensure that during inference we can deal with different split as well. We will include a reference to the query training.
> 3. $I(\cdot)$ is the indicator function to ensure that the output is a probabilistic vector. We will clarify this in the revision.
> 4. We use $\alpha=0.125$ throughout the experiment. In general, one can heuristically pick $\alpha$: for example, starting with a big $\alpha$, and halve whenever the maximum number of allowed iterations is reached.
> 5. 92.95$\%$ is the classification accuracy with $40\%$ missing pixels.

---

> > ### Comment · Reviewer_8YQo · 2021-11-30
> > **Unchanged score.**
> >
> > Thanks for your rebuttal and trying the Baseline 1 that I suggested. The results of that experiment are in line with my expectations: The monotonicity constraint, while theoretically appealing, does not result in a practical advantage. I think it would be useful to find a use case where this monotonicity results in a practical advantage (and I believe this might be hard to find - which severely limits applicability).

---

### Official Review · Reviewer_1tpT · 2021-11-03

**Correctness:** 4
**Technical Novelty And Significance:** 3
**Empirical Novelty And Significance:** Not applicable
**Recommendation:** 6
**Confidence:** 4

**Main Review:**

On one hand, this paper has some significant strengths. First, the paper is fairly well written in general. Second, while this work is heavily inspired by Winston & Kolter (2020), I find that the connection between mean field and monotone DEQ is quite interesting (although relatively straightforward), and the proposed method is theoretically well founded.

On the other hand, the paper also has some limitations.

1. First and foremost, I find the experiments quite limited, which is also acknowledged by the authors. A more diverse set of applications would have made the paper much more solid. At the very least, I would have expected some experimental comparison with restricted Boltzmann machines (not to mention also its variants such as extensions to multi-label). The proposed model is theoretically sound, but it is not clear why one should use it.


2. The paper also has some minor presentation issues, but before ending my review with them, I would like to have some comments on the bibliographical discussion.

2a. Since the convergence of mean field is presented as an emphasis in the paper, I would like to point out a very recent NeurIPS 2021 paper on the topic: "Regularized Frank-Wolfe for Dense CRFs: Generalizing Mean Field and Beyond" (https://arxiv.org/abs/2110.14759). In this paper they view parallel mean field as an instance of the generalized conditional gradient method and thus obtain different convergent variants of parallel mean field with different step-size rules. It seems to me that these variants do not have the same limitations as Krahenbuhl's and Baqué's as discussed in this paper (even though their resulting algorithms seem to be similar to Baqué's at first glance). Could you give some comments on this? Including such discussion would give the reader a broader and more up-to-date view of the current state of the art. (Of course no experimental comparison would be needed, that's not the focus of the paper).


2b. "Numerous works also try to combine deep neural networks with conditional random fields (CRF) (Arnab et al., 2018; Schwartz et al., 2017; Zheng et al., 2015)."

Even though this is just a minor detail in the current paper, I would like to take this opportunity to raise an important issue regarding credit assignment.

The first to view "CRFs as RNNs" for the dense CRFs of Krahenbuhl & Koltun (2011) was actually Krahenbuhl & Koltun (2013) and not Zheng et al. (2015). Krahenbuhl & Koltun (2013) had two major contributions in their paper: (a) convergent parallel mean field, and (b) parameter learning of dense CRFs with reverse-mode automatic differentiation (i.e., viewing "CRFs as RNNs" and backpropagating through time). Unfortunately, Krahenbuhl & Koltun (2013) have been often credited with (a) only and not (b), while (b) is to me even more significant than (a). This is not fair, and I think this happened because some previous work didn't cite them correctly or in a misleading manner. For example, Arnab et al., (2018) didn't even cite this paper (even though they did cite in their previous work (Zheng et al., 2015), not sure why they removed the citation from the journal version). The fact that Arnab et al., (2018) completely ignored Krahenbuhl & Koltun (2013) and credited Zheng et al., (2015) for viewing "CRFs as RNNs" made their presentation misleading (and unacceptable to me).

I would like to encourage the authors to give proper credits to Krahenbuhl & Koltun (2013) whenever they have an opportunity to do so. Starting with the current submission, I would suggest to slightly change the above sentence to the following, for example:

"Numerous works also try to combine conditional random fields (CRF) with pixel-wise classifiers (such as neural networks) to obtain fully end-to-end models (Krahenbuhl & Koltun, 2013; Schwartz et al., 2017; Zheng et al., 2015)."

But of course it is up to the authors to decide.


3. Some comments on the presentation:

Major:

In the abstract: "In addition, we show that our procedure outperforms existing mean-field approximation methods while avoiding any issue of local optima." I guess the authors are referring to the comparison with Krahenbuhl's and Baqué's that is presented in the appendix. If something is mentioned in the abstract, then it's important enough to be included in the main content instead of being left in the appendix. I would suggest to either remove the above sentence from the abstract, or to move such comparison from the appendix to the main content (the former seems more appropriate to me, since this is not the focus of the paper).


Minor:

- Eq. (1) should end with a comma instead of a dot.

- Page 3, 1st paragraph: "proposed a deep parameterization of MRF. However, their..." -->

- Page 3, 1st paragraph: "proposed a deep parameterization of MRF, but their..."

- Section 3.1, 1st paragraph: lines 3-4 are not clear to me.

- Page 6, before Eq. (13): "similarly-factored A matrix" --> "similarly-factored matrix A"

**Summary Of The Paper:**

This paper proposes a class of model called monotone deep Boltzmann machines, where the underlying potentials are parameterized (e.g., by CNNs) such that they obey some monotonicity constraint. This constraint ensures that the inference problem has a global optimum, which can be found using some generalized variant of parallel mean field. The method is inspired from monotone DEQ, previously proposed by Winston & Kolter (2020). Experiments on a joint task of image denoising and classification show that the proposed method can effectively model complex data distributions such as images.

**Summary Of The Review:**

Interesting and theoretically sound model. The set of experiments is quite modest, in addition to some minor presentation issues.

---

> ### Author Response · Authors · 2021-11-14
> **Thanks for your review**
>
> We appreciate the valuable comments by the reviewer. [1] is definitely a relevant reference that we will mention in the revision. Their Frank-Wolfe (FW) update is slightly different from Baque's update, as the FW damping happens outside the nonlinearity and they directly damp the probability vector; while Baque's method has damping inside the nonlinearity, and they damp the log-probability vector. The FW paper asserts their algorithm converges to a stationary point, which may or may not be the global minimum (since the energy function is quadratic, and even with the monotonicity condition it may not be convex), while our algorithm asserts global convergence to global minimum. Note that the FW convergence is also global, meaning the algorithm will converge no matter where the initial guess is. This is still a step further to Baque's analysis, where they only show the loss function keeps decreasing at every step. We can numerically show that the FW update may not converge to the mean-field fixed point, and will add the comparison to the revision.
>
> We will also give proper credits to Krahenbuhl & Koltun (2013) for the CRFs as RNN view, and reorganize our presentation so that the abstract and main text match.
>
> Regarding experimental comparisons, we are working on comparisons to RBMs, which has been requested by several reviewers.
>
>
> [1]: Regularized Frank-Wolfe for Dense CRFs: Generalizing Mean Field and Beyond

---

> > ### Comment · Reviewer_1tpT · 2021-11-15
> > **Thanks for the feedback**
> >
> > Dear authors,
> >
> > Thank you for your reply.
> >
> > In my opinion, experiments with FW should not be that important to your paper (even though they are "nice to have"). I would recommend to focus on RBM (which should have the highest priority) and the other reviewers' concerns.
> >
> > Looking forward to reading your revised manuscript.
> >
> > Best regards.

---

> > > ### Author Response · Authors · 2021-11-23
> > > **Revision**
> > >
> > > Thanks for your update. We have submitted a revision and included a description in the **Summary of Revision** response.

---

> > > > ### Comment · Reviewer_1tpT · 2021-11-23
> > > > **Thanks but none of my concerns was addressed as promised**
> > > >
> > > > Dear authors,
> > > >
> > > > I've skimmed quickly the latest revised manuscript but it doesn't seem to have addressed **any** of my concerns and suggestions (which you promised to do), including: (1) discussion on convergent MF based on Frank-Wolfe (no experiments required), (2) proper credits to Krahenbuhl & Koltun (2013) for viewing "CRFs as RNNs", and (3) experimental comparison with RBM.
> > > >
> > > > Am I missing something? You have uploaded the wrong PDF maybe?
> > > >
> > > > Thanks.

---

> > > > > ### Author Response · Authors · 2021-11-23
> > > > > **Sorry about the confusion**
> > > > >
> > > > > We have reuploaded a revision and highlighted the major changes using blue color. We put the RBM experiments in the appendix for now to meet the page limit. Thanks.

---

### Author Response · Authors · 2021-11-22
**Summary of Revision**

We thank all the reviewers for their reviews. One common baseline brought up by many reviewers is the RBM/DBM. We conduct the same joint MNIST imputation-classification experiment using DBM, which has 2 hidden layers: the first hidden layer has 300 units and the last hidden layer has 10 units. The input visible layer has 784 units. In total, this amounts to 238,200 parameters (our proposed model uses 165,300 parameters in the MNIST experiments). The test accuracy using DBM with 60% random pixels masked off is 93.58%. Our model achieves comparable test accuracy (92.95%) and imputation compared to this DBM, given fewer parameters and despite the monotonicity constraint of our model. See more details in the revised version, appendix C.

---

### Decision · Program_Chairs · 2022-01-20

**Decision:**

Reject

**Comment:**

This is an interesting contribution to the Boltzmann machine (BM) literature that makes a nice connection to DEQ models. On a positive note, reviewers found that it was well-written, clear, and interesting. Unfortunately, there were significant concerns with the manuscript that were not fully addressed in the revision: inappropriate or incomplete baselines, insufficient credit given to previous works, and the fact that this model is limited as compared to its BM relatives.

I would recommend that the authors take into account the reviewers' feedback in a revision of the work.